# Treatment Pathways and Prognosis in Advanced Sarcoma with Peritoneal Sarcomatosis

**DOI:** 10.3390/cancers15041340

**Published:** 2023-02-20

**Authors:** Fabian Klingler, Hany Ashmawy, Lena Häberle, Irene Esposito, Lars Schimmöller, Wolfram Trudo Knoefel, Andreas Krieg

**Affiliations:** 1Department of Surgery (A), Heinrich-Heine-University and University Hospital Duesseldorf, D-40225 Duesseldorf, Germany; 2Institute of Pathology, Heinrich-Heine-University and University Hospital Duesseldorf, D-40225 Duesseldorf, Germany; 3Department of Diagnostic and Interventional Radiology, Medical Faculty, University Duesseldorf, D-40225 Duesseldorf, Germany

**Keywords:** sarcomatosis, sarcoma, peritoneal

## Abstract

**Simple Summary:**

Presentation of sarcoma inside the peritoneal cavity is a rare finding to begin with. In such a rare incidence, there are a multitude of sarcoma subtypes that can be identified, with each of these subtypes presenting with different characteristics in terms of prognosis and treatment options. Considering these factors and the resulting lack of strong data to guide treatment plans, this study aims to share our experiences with cases of peritoneal sarcomatosis to increase the knowledge about possible options and outcomes. We report on 19 cases of surgery in patients with peritoneal sarcomatosis, ranging from palliative procedures to major multivisceral resections, and highlight their course of disease, treatment, and outcome. Hereby, we aspire to increase the cumulative experience with challenging cases like these and support a more informed tailoring of treatment plans for future cases to come.

**Abstract:**

Sarcomas represent a heterogeneous group of mesenchymal malignancies that most commonly occur in the extremities, retroperitoneum, and head and neck. Intra-abdominal manifestations are rare and prove particularly difficult to treat when peritoneal sarcomatosis is present. Because of the overall poor prognosis of the disease, a tailored approach to surgical management is essential to achieve satisfactory outcomes with limited morbidity. We present the perioperative and long-term outcomes of 19 cases of sarcoma with peritoneal sarcomatosis treated surgically at our hospital. Treatment pathways were reviewed and clinical follow-up was performed. Patient characteristics, medical history, tumor subtype, surgical approach, hospital stay, complications, follow-up, and overall survival (OS) were assessed. Our patients were 9 women and 10 men with a median age of 45.9 years (18–88) and a median survival of 30 months (0–200). In most cases, peritoneal sarcomatosis was either discovered during surgery or the procedure was performed with palliative intent from the beginning. The surgical approach in these cases is very heterogeneous and should consider a variety of factors to tailor an approach for each patient. Sharing our experiences will help to increase knowledge about this rare disease and provide insight into the management of future cases.

## 1. Introduction

Soft tissue sarcomas (STSs) account for less than 1% of malignancies and are thus a rare entity [1]. In industrialized nations, STSs occur with an incidence of 4–5/100,000 residents per year. Owing to their wide distribution in localization and with more than 80 further specified histological subtypes, they comprise a very heterogeneous group of malignancies [1,2]. However, continuous efforts to characterize and subclassify STSs, taking molecular pathology into account, proves to be the most promising approach for individualized treatment efforts [3,4]. The growing understanding of the molecular differences between tumors that used to be viewed as indiscriminative offers great chances for future efforts to characterize these subtypes clinically and develop a more customized treatment approach. Previous attempts to develop standardized treatment protocols have faced difficulties in addressing all of the individual requirements and it is challenging to obtain reliable data for a specific subtype. Especially in advanced cases with recurrent or metastasized disease, finding the fine line between undertreatment and overtreatment can be a huge challenge. STS with the involvement of the peritoneum is even more uncommon than other sites of metastasis, hence there is very limited experience from which to draw [5]. Therefore, evidence-based treatment for these cases has not yet been established.

Treatment options for patients with peritoneal sarcomatosis (PS) range from palliative treatment to aggressive systemic therapy and include extensive cytoreductive surgery (CRS) and hyperthermic intraperitoneal chemotherapy (HIPEC) in selected patients. Given the high morbidity and differential survival benefit of cytoreduction and HIPEC, the approach to PS remains controversial [6,7]. As a result of the lack of sufficient data, current recommendations for treatment pathways mostly consist of consensus statements [8]. While most studies focus on a single histologic subtype or attempt to bundle sarcoma patients as a whole, there is a distinct lack of data focusing on advanced stages such as PS. In addition, to the best of our knowledge, there is no study that has focused on the role of surgery as a palliative modality in PS. Thus, the aim of our study was to present cases with PS that were treated at our department in order to illustrate the wide variety of therapeutic options available in the development of a tailored approach for each individual patient and to provide examples for future treatment plans. We further aim to showcase the highly heterogeneous outcome for patients with different STSs and PS to encourage continuous efforts to develop evidence-based treatment protocols.

## 2. Materials and Methods

Of the 291 sarcoma cases we have treated since September 2003, we identified 19 patients who underwent surgical procedures for PS. In all cases, an individualized approach was chosen depending on the disease course, prognosis, and patient requirements. We conducted a preoperative interdisciplinary discussion in our tumor board specialized in sarcomas, in which recommendations for the therapeutic regimen were made, considering histology, clinical findings, physical performance, and patient preferences.

Patients were selected from a retrospectively collected and prospectively maintained database of sarcoma patients treated at our university hospital. For the study, we included only surgically treated patients with histologically confirmed PS either before or at the time of their surgical procedure. We obtained approval for this study from the ethics committees of the Medical Faculty of Heinrich Heine University, Düsseldorf (study number: 2022–2010).

The following information was obtained from the patients’ medical records: age at surgical intervention, sex, time between initial diagnosis and diagnosis of PS, disease status at presentation (primary or recurrent), prior treatment, change of treatment centre, tumor subtype, details of treatment (palliative or curative intent, resected structures, surgical margins, reconstructive technique, Clavien–Dindo surgical complications, length of hospital stay), and follow-up.

Follow-up and survival times after discovery of PS and after primary diagnosis of sarcoma were calculated and corresponding Kaplan–Meier survival curves were generated. The statistical analyses were conducted with the software R (version 1.4.1106) utilizing the packages readxl, survminer, and survival [9,10,11].

## 3. Results

### 3.1. Patient Characteristics

Detailed patient characteristics are summarized in Appendix A. The median age at the time of PS diagnosis was 45.9 years, with large differences between tumor types. Nine of the patients were female and ten were male. Among the 19 patients we identified, there were 8 different tumor entities (Table 1).

### 3.2. State of Disease

Twelve of the patients were found to have a primary tumor in the abdominal cavity. Four patients had PS at the time of their primary diagnosis, with ten patients treated for more than 12 months before sarcomatosis occurred. The median time between initial diagnosis and PS was 16.2 months. We noted that 11 of the patients had been treated for their disease at another institution before being treated at our hospital. Thirteen of the patients had already undergone surgical resection prior to PS diagnosis and, in eight of these patients, the tumors were not completely removed from healthy tissue microscopically or macroscopically. In two cases, the primary tumor originated from the retroperitoneal space, and in both cases, R1/R2 resection was documented before the discovery of PS. In ten cases, distant metastases were already present at the time of PS discovery (Table 2, Figure 1).

### 3.3. Treatment

Each patient underwent surgical resection of the tumor mass at some point, with an average of 3.8 resections during the course of the disease (Table 3). The list of operations ranged from diagnostic procedures and bypass surgery to metastasectomy and major multiorgan resections (Figure 2).

At the time of diagnosis of PS, we treated nine patients with curative intent, sometimes including resection of the symptomatic tumor mass, and at other times to prevent future complications. In two cases, palliative treatment was chosen at another institution and resection with curative intent was performed after referral to our hospital.

Four patients received CRS and cisplatin-based HIPEC for PS, all of whom had desmoplastic round cell tumors (DSRCTs).

### 3.4. Follow-Up

The mean follow-up was 22 months ranging from 0 to 172 months (Table 4). Six patients achieved a survival of greater than 5 years after primary diagnosis of sarcoma, while five patients are still alive at the time of publication of this study. The median follow-up of these surviving patients was 13 months, ranging from 2 to 172 months. Complications were common in these patients, with only five patients without any surgical complications. Seven patients developed wound healing issues and two patients died from terminal respiratory insufficiency after a complicative postoperative course.

The patients we treated with curative intent were on average 40 years old, in contrast to an average age of 55 years for patients who received palliative surgery. Survival rates differed significantly between the groups and showed a longer overall survival after primary diagnosis as well as after diagnosis of PS for patients treated with a curative intent (Figure 3). Interestingly, patients receiving curative therapy tended to have fewer severe complications. In contrast to the group of patients who underwent curative surgery, all of whom survived the hospital stay, we observed an in-hospital mortality rate of 20% in patients who underwent palliative surgery. However, hospital stay did not differ between the two groups (Table 5).

The median overall survival after primary diagnosis of sarcoma was 76 months in patients treated with a curative intent versus 22 months in patients with a palliative treatment plan, and the 5-year survival rates were 52% versus 23%. After diagnosis of PS, the median survival was 30 months and 7.5 months when comparing curative and palliative treated patients, respectively, with a 5-year survival rate of 40.2% versus 13%, respectively.

## 4. Discussion

Surgery with complete en bloc resection of all adjacent tissues and organs, combined with or without radiation therapy, remains the primary and only truly curative treatment option for localized and clinically resectable STSs. It is also recommended for advanced or metastatic STSs. In this context, factors such as isolated oligometastatic disease, long disease-free interval, favorable histology, response to chemotherapy, and high likelihood of complete resection make the argument for surgery even in advanced and recurrent disease [12]. Interestingly, a recently published meta-analysis also suggests that CRS with HIPEC may improve prognosis in a selected group of PS patients [13]. In addition, patients with advanced, primarily inoperable STS may also be offered palliative surgery for symptom control of tumor-related complications such as pain, bleeding, or bowel obstruction [14]. While the adjuvant and neoadjuvant treatment of patients with STS is still controversial and no standardized regimen exists, anthracycline-based chemotherapy in combination with ifosfamide is used as a first-line therapy in the treatment of advanced STS [15,16]. A study by Gough et al. demonstrated that palliative chemotherapy with doxorubicin alone or in combination with ifosfamide, as well as combination chemotherapy with gemcitabine and docetaxel, significantly reduced pain and sleep disturbance while worsening fatigue [17]. However, not all patients with advanced or metastatic STS benefit from conventional chemotherapy, and targeted therapy may play the most important role in the treatment of patients who are resistant to conventional chemotherapy or in whom conventional chemotherapy has failed. Accordingly, in recent years, increasing numbers of preclinical studies have been conducted to explore the pathogenesis and potential therapeutic targets of STS [4,18,19], and clinical trials have been initiated to target different molecules in distinct histologic subtypes [20], hopefully opening new doors in the clinical management of patients with advanced or metastatic disease in the future. Although the proportion of patients receiving outpatient palliative care for STS is likely to be very small [21], specialized palliative care interventions have recently been shown to result in significant symptom relief in patients with advanced STS, and early integration of palliative care in these patients is thus recommended [22]. Nevertheless, the treatment decision should be made in the context of a multidisciplinary discussion.

However, treatment recommendations for sarcoma patients increasingly depend on their individual tumor subtype [23], while scientific advances have provided increasing rationale for differentiating STSs with a focus on genetics and potential molecular targets for individualized treatment options. The increasing distinction of subtypes with different biological and clinical aspects makes it difficult to draw conclusions for individualized treatment pathways for each sarcoma subtype, especially for smaller studies [24]. As we were confronted with the same problem in our study, we aimed to provide a largely descriptive overview of our experience with PS. Because one-third of our patients are the only patients with their respective histologic subtype, we cannot claim to present conclusive evidence of standardized treatment. Nevertheless, we believe that any contribution to the collective knowledge of these difficult cases is valuable.

The highest number of patients with a tumor subtype in our study was generated by pleomorphic sarcomas—not otherwise specified (NOS), formerly known as malignant fibrous histiocytomas. This group accounts for 10–15% of STSs and is typical of low-differentiated tumors prone to aggressiveness and early metastasis. It is diagnosed less frequently today than twenty years ago, as technological improvements often make it possible to determine a line of differentiation [25]. Nevertheless, the group remains inhomogeneous in terms of age, tumor location, disease progression, and overall survival. Surgical intent was mostly palliative and complication rates were high. One case stood out with significantly higher overall survival—in this case, the primary diagnosis was sarcoma from a morcellated uterus during hysterectomy. We can only speculate whether the mere location of the primary tumor played a role in the favourable outcome or whether there was an affiliation with uterine leiomyosarcomas that could not be determined. Nevertheless, further progress in determining a histologic subtype of these low-differentiated tumors should be helpful in identifying the ideal treatment for these patients.

One of the more uniform patient groups in our study was those with DSRCTs. As DSRCTs typically occur in adolescent patients, we found our youngest patients in this group [26]. These often physically fit and motivated patients are dealing with a very aggressive disease with a 5-year overall survival rate of 15% to 30% [26,27]. In light of this, aggressive interdisciplinary treatment with perioperative chemotherapy following Ewing protocol and radical surgery with peritonectomy offers the best chance for patients [28]. As most cases of DSRCTs have peritoneal seeding, there is more information about PS in this type of sarcoma. Additional HIPEC after CRS is being investigated in DSRCTs and seems to provide additional benefit, which is why these patients were the only ones to receive this treatment in this study [29]. Other treatment options such as radiotherapy, targeted therapies, or the use of regional deep hyperthermia protocols should be discussed in serial tumor board meetings [26,30]. The patients with DSRCTs that we treated received aggressive treatment with multivisceral resection and HIPEC, often with repeated surgeries during the course of their disease, but incredible recovery and only minor complications were demonstrated in three of four patients.

Differentiating the extent of peritoneal involvement of a malignancy is an established concept for peritoneal carcinomatosis. The peritoneal carcinomatosis index (PCI) has been widely used for prognostic reasons or to evaluate aggressive treatment options such as cytoreductive surgery (CRS) and HIPEC [31,32]. The concept has also been used for PS, but its prognostic value for a benefit of CRS and HIPEC in PS remains controversial [33,34,35]. If future studies succeed in defining indications for CRS and HIPEC outside of DSRCTs, PCI is likely to become more important in the treatment of peritoneal metastatic sarcomas.

With more than 20% of all STSs in Germany, liposarcoma occupies a significant role in this group of rare diseases [1]. Even though there are more data available in relation to other STS types, scientific approaches to understand PS in liposarcoma are just beginning to take shape [5]. Well-differentiated liposarcomas have low metastatic rates, whereas dedifferentiated liposarcomas are more aggressive and are associated with a worse prognosis. Myxoid liposarcomas account for approximately 20–30% of all liposarcomas and typically carry a genetic translocation resulting in a higher rate of primary multifocal appearance and metastasis, but still have a significantly better overall survival rate than dedifferentiated liposarcomas [1,4,36]. Our results support these data, as we only studied PS in dedifferentiated and myxoid liposarcomas. The two patients we treated for myxoid liposarcoma both underwent multiple surgical resections, received various chemotherapies, and had an overall survival of more than 50 months. In contrast, the overall survival of the patients we treated for dedifferentiated liposarcoma was much lower, the rate of distant metastases was high, and recurrence was rapid. Looking at one of the patients we treated with palliative small bowel bypass anastomosis for tumor-associated ileus, we managed to discharge the patient without surgical complications, but unfortunately, he died the following month.

Leiomyosarcoma accounts for approximately 10–25% of STSs in Germany [1]. Both patients had a history of hysterectomy, making uterine primary disease likely, although histologic evidence of uterine tumor was obtained in only one case. Five-year survival rates for uterine leiomyosarcomas are 20–60%, which compares favourably with many of the previously mentioned tumor types, and our data support these findings with overall survival since primary diagnosis of 205 and 97 months, respectively, in our patients, both of whom are alive to date. It is worth noting that both patients received multiple surgical resections and systemic therapies during the course of their disease, but achieved long periods of tumor control. In one case, the patient was considered palliative owing to recurrent locally advanced disease, but opted for surgical resection with local peritonectomy. The patient received perioperative chemotherapy with a regional deep hyperthermia protocol and, to date, has been disease-free for 14 years without further treatment. Uterine leiomyosarcoma appears to reward persistent efforts to control the disease and offers opportunities for successful systemic and surgical treatment even with advanced disease.

The patient we treated for MPNST was in the expected age group for patients with neurofibromatosis type 1, and because MPNST is a highly malignant sarcoma, tumor recurrence and distant metastases ensured a rapid disease progression [1].

SFT is a subtype with a low metastatic burden, but also low sensitivity to chemotherapy, so the focus is on local control of the tumor [37]. We treated the patient with total parietal peritonectomy, right hemicolectomy with resection of the terminal ileum, radical lymphadenectomy, and partial resection of the urinary bladder. Although the postoperative course was not without complications, the patient is still alive and well 41 months later.

In one case, we treated a patient with advanced osteosarcoma with peritoneal involvement. Intra-abdominal metastases in osteosarcoma are very rare and often occur late in the disease course. A recent study of abdominally metastatic osteosarcomas found that 75% of patients died within 6 months of diagnosis of peritoneal involvement [38]. Nevertheless, resectable disease can be treated surgically and may be beneficial to patients. In our case, the patient could be discharged after 17 days of hospitalization and lived for 9 months after extensive abdominal surgery.

The tumor site from which sarcoma originated varied widely between cases and did not appear to play a role once PS was detected. Two of the documented cases started in the retroperitoneal space. In both cases, there was a history of previous incomplete resection prior to the development of PS. This supports the existing view that a clean surgical technique with clear resection margins is one of the most important factors in avoiding PS in surgical treatment of retroperitoneal STS [39].

Because most of our patients received surgical treatment for their malignancy before a diagnosis of PS, it is not surprising that a high number of R1/R2 resections lead to PS. Our data underscore the principle of negative surgical margins, supporting one of the well-established principles in sarcoma treatment [40,41].

During data collection, we found that most patients were referred to more than one centre during the course of their disease and that treatment plans changed frequently when the physician changed. In fact, a study from France demonstrated that more than 40% of histologic diagnoses were changed after obtaining a second opinion [42]. Numerous data suggest that patients benefit from treatment in a larger institution where multidisciplinary tumor boards discuss therapeutic options [43,44].

By focusing on advanced-stage sarcoma with PS, we hope to provide valuable experience for surgeons trying to find the right treatment path for their patients, taking into account the disease course and histologic subtype of the sarcoma. In contrast to other studies that focus on curative therapy for PS, we also investigated the importance of palliative surgery in this patient population. Importantly, to the best of our knowledge, there are no studies specifically evaluating the value of palliative surgery in patients with PS. However, there are a limited number of studies that focus exclusively on the quality of life and oncologic benefit of palliative resection for retroperitoneal or intra-abdominal STSs. In the study by Yeh and co-workers, in a collective of 1084 patients with intra-abdominal STS, palliative procedures were performed in 112 patients and were surgically performed in 82% of cases [45]. Of these, palliative procedures were most commonly undertaken in the gastrointestinal tract (44%). Symptom burden improved in 71% of patients 30 days after surgery, while only 54% of patients were symptom-free at 100 days. Moreover, 54% of obstructive gastrointestinal symptoms were successfully resolved at 30 days and 23% of patients were also symptom-free at 100 days. While the overall morbidity in this study was 29%, the postoperative mortality was 12%. A study by the U.S. Sarcoma Collaborative analyzed the results of palliative resection in 70 patients with retroperitoneal STS [46]. The predominant indication for palliative surgery was pain or bowel obstruction. However, the authors also observed a relatively high morbidity rate of 38% in their study. Unfortunately, these relatively high morbidity and mortality rates are consistent with our results, as we also observed a mortality rate of 20% and a major complication rate of 40% in the palliative surgery group in our own cohort of patients with PS. While it may be tempting to treat an obvious surgical emergency, it can be devastating for the patient to spend valuable time recovering from surgery or its complications while the disease inexorably progresses. Therefore, although palliative surgery for PS may improve symptom burden in these patients, given the limited oncologic benefit and increased postoperative morbidity and mortality, surgical therapy for symptom control should be considered only after careful selection and risk–benefit analysis in specialized centers.

## 5. Conclusions

Overall, patients with PS have a high rate of surgical complications and poor survival rates, and indications for further surgical intervention should take this into account. It should be noted that, in many cases, PS could be detected only during surgery. Nevertheless, in selected patients, considering tumor subtype, physical performance, and patient preference, surgical resection may be beneficial even if PS is detected.

The individualization of a treatment pathway always has the potential to be superior to a standardized approach by providing additional information. However, if the information available is based on unrepresentative personal experience or is not tailored to the specific case, there is also a risk of over- or undertreatment of the patient. Sharing experiences with colleagues, making treatment recommendations based on scientific evidence and consensus, and discussing all options with the patient are essential in the management of PS.

In the future, the use of large STS databases to generate reliable data on individual STS subtypes and stages of progression would be necessary to allow a scientific-based tailored treatment plan for difficult cases such as these. In our view, however, surgical control remains crucial in patients with PS and warrants a greater level of effort.

## Figures and Tables

**Figure 1 cancers-15-01340-f001:**
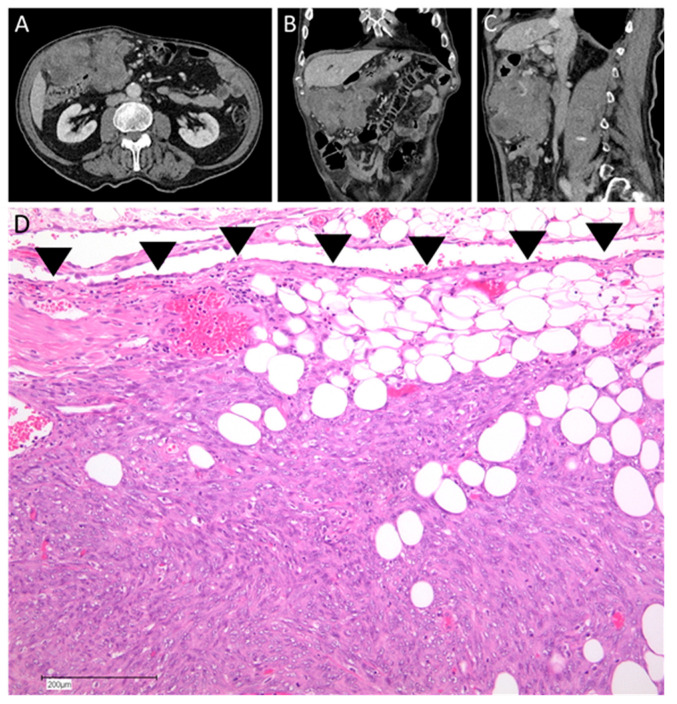
Example for diagnosis of a peritoneal metastasized soft tissue sarcoma. The patient was an 87-year-old male who presented with a palpable growing mass in his abdomen. (**A**–**C**) Abdominal CT-scan showed a large tumor in his right upper quadrant and suggested infiltration of the tumor into the distal stomach and ascending colon. (**D**) Histophotograph of the tumor shows a neoplasm consisting of atypical spindle cells infiltrating the peritoneum (arrowheads: mesothelial lining) (100×, H&E). Histology report showed a high-grade pleomorphic sarcoma—not otherwise specified (NOS).

**Figure 2 cancers-15-01340-f002:**
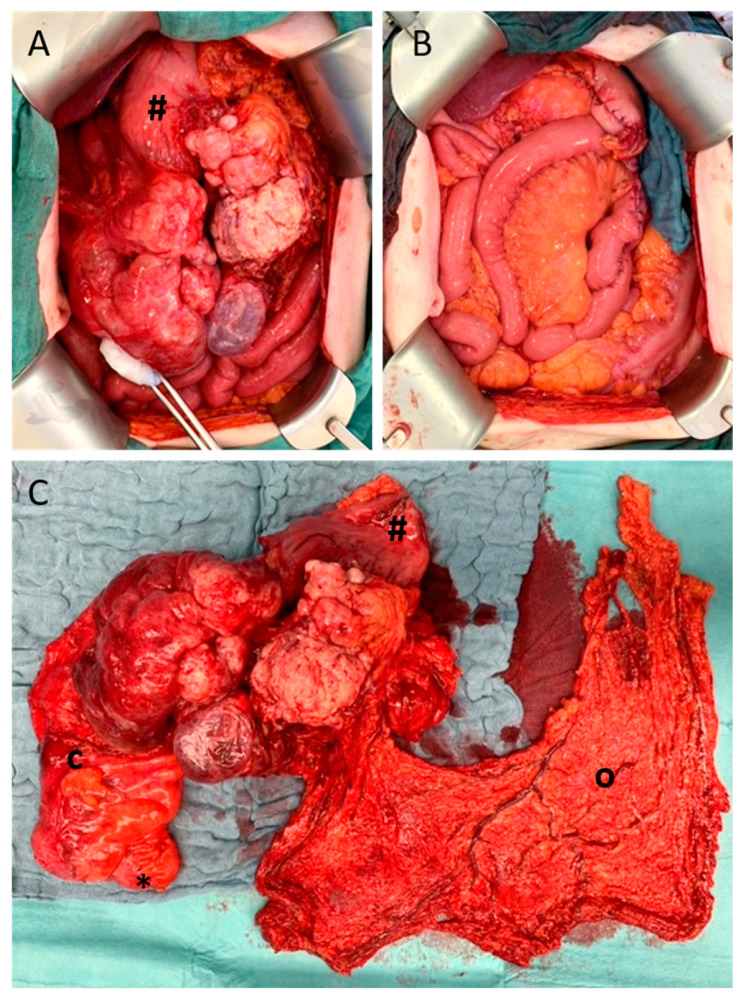
Example of palliative resection. Intraoperative findings of the previously mentioned 87-year-old male with pleomorphic sarcoma (Figure 1). (**A**) Surgical exploration showed diffuse peritoneal sarcomatosis and a massive tumor formation. (**B**) Palliative resection and reconstruction with gastrojejunostomy, Roux–en–Y reconstruction, and terminal ileostomy were performed, and the patient was discharged three weeks later. (**C**) The resected specimen contained the tumor with the infiltrated distal stomach (#) and ascending colon (**c**) and terminal ileum (*), as well as the greater omentum (**o**).

**Figure 3 cancers-15-01340-f003:**
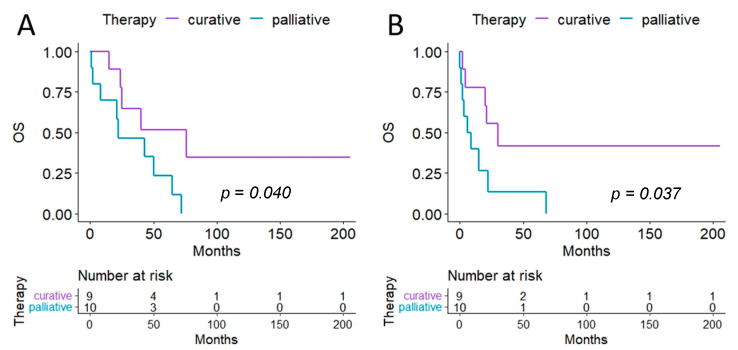
Overall survival. (**A**) Kaplan–Meier survival curve after primary diagnosis of sarcoma is shown. (**B**) These Kaplan–Meier curves show survival after the diagnosis of peritoneal sarcomatosis has been made.

**Table 1 cancers-15-01340-t001:** Tumor subtypes and patient characteristics.

Tumor Subtype	Number of Patients	Median Age (Range)	Sex Male/Female
NOS	5	67 (46–88)	3/2
DSRCT	4	18 (18–24)	2/2
Dedifferentiated liposarcoma	3	65 (36–84)	1/2
Myxoid liposarcoma	2	61 (44–64)	2/0
Leiomyosarcoma	2	51 (44–58)	0/2
MPNST	1	24	0/1
SFT	1	68	1/0
Osteosarcoma	1	26	1/0
**Total**	**19**	**45.9 (18–88)**	**10/9**

Desmoplastic small round cell tumor (DSRCT), pleomorphic sarcomas—not otherwise specified (NOS), malignant peripheral nerve sheath tumor (MPNST), solitary fibrous tumor (SFT).

**Table 2 cancers-15-01340-t002:** State of disease at discovery of peritoneal sarcomatosis.

Tumor Subtype	Number of Patients	Mean Time Since Primary Diagnosis (SD)	Prior External Treatment	Mean Prior Resections (Range)	Prior R1/R2	Distant Metastasis
NOS	5	18 (26.1)	40%	3.4 (0–10)	60%	20%
DSRCT	4	6 (9.6)	75%	0.3 (0–1)	25%	50%
Dedifferentiated liposarcoma	3	7 (10)	33%	1.7 (0–4)	66%	100%
Myxoid liposarcoma	2	40 (7.4)	100%	1.5 (1–2)	0%	100%
Leiomyosarcoma	2	9 (11.8)	100%	1.5 (1–2)	ND	50%
MPNST	1	13	100%	3	100%	100%
SFT	1	27	0%	1	0%	0%
Osteosarcoma	1	34	0%	2	0%	100%
**Total**	**19**	**16.2 (17.94)**	**57.9%**	**1.8**	**42.1%**	**52.6%**

Peritoneal sarcomatosis (PS), standard deviation (SD), desmoplastic small round cell tumor (DSRCT), pleomorphic sarcomas—not otherwise specified (NOS), malignant peripheral nerve sheath tumor (MPNST), solitary fibrous tumor (SFT), not defined (ND).

**Table 3 cancers-15-01340-t003:** Treatment.

Tumor Subtype	Number of Patients	Total Number of Resections per Patient	Surgical Treatment Spectrum			
			Exploration and Biopsy	Palliative Procedure	Limited Resection	Multivisceral Resection
NOS	5	5			15	11
DSRCT	4	2.75	2		1	7
Dedifferentiated liposarcoma	3	2	2	1	2	3
Myxoid liposarcoma	2	5		1	3	6
Leiomyosarcoma	2	6	2		5	6
MPNST	1	4			2	2
SFT	1	2	1		1	1
Osteosarcoma	1	3			2	1
**Total**	**19**	**3.8**	**7**	**2**	**31**	**37**

Desmoplastic small round cell tumor (DSRCT), pleomorphic sarcomas—not otherwise specified (NOS), malignant peripheral nerve sheath tumor (MPNST), solitary fibrous tumor (SFT).

**Table 4 cancers-15-01340-t004:** Follow-up.

Tumor Subtype	Number of Patients	Mean Survival in Months Since Primary Diagnosis (SD)	Mean Survival in Months after PS	Major Complications (Clavien–Dindo 3 or 4)	Mean Follow-Up in Months (SD)
NOS	5	38 (31.2)	20 (27.8)	60%	17 (25.3)
DSRCT	4	24 (1.0)	17 (9.2)	25%	13 (7.4)
Dedifferentiated liposarcoma	3	10 (10.3)	3 (2.5)	33%	3 (2.3)
Myxoid liposarcoma	2	63 (18.3)	22 (10.9)	0%	20 (9.6)
Leiomyosarcoma	2	151 (76.5)	143 (88.3)	0%	115 (80.4)
MPNST	1	15	2	100%	2
SFT	1	62	36	100%	3
Osteosarcoma	1	43	9	0%	8
**Total**	**19**	**46 (47.4)**	**30 (47.9)**	**31.6%**	**22 (40.2)**

Peritoneal sarcomatosis (PS), standard deviation (SD), desmoplastic small round cell tumor (DSRCT), pleomorphic sarcomas—not otherwise specified (NOS), malignant peripheral nerve sheath tumor (MPNST), solitary fibrous tumor (SFT), standard deviation (SD).

**Table 5 cancers-15-01340-t005:** Treatment intention and outcome.

	Number of Patients	Age in Years	Major Complications (Clavien–Dindo 3 or 4)	Hospital Stay in Days (SD)
Curative Intention	9	40	22%	29 (16)
Palliative Intention	10	55	40%	27 (21)

Peritoneal sarcomatosis (PS), standard deviation (SD).

## Data Availability

The data presented in this study are available upon request from the corresponding author.

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
