# Peer review of "Treatment Pathways and Prognosis in Advanced Sarcoma with Peritoneal Sarcomatosis"

_cancers, 2023, doi:10.3390/cancers15041340_

Round 1
Reviewer 1 Report
This article provides an in-depth analysis of the treatment pathways and prognosis of advanced sarcoma with peritoneal sarcomatosis. The authors present a comprehensive overview and observation of the clinical evidence. The article is well-written and clearly laid out, making it easy to follow the complex information presented. It is an invaluable resource for clinicians working with advanced sarcoma patients.
Minor revisions to be carried out.
Discuss the potential benefits and drawbacks of different surgical and medical treatments, as well as the role of palliative care.
Thorough discussion of the most appropriate treatment strategies should be discussed.
The results should also be discussed in the context of clinical approaches for other types of cancer therapeutics referring to latest articles (PMID: 36149747, PMID: 29494958)
Author Response
We thank you for the review of our manuscript and the valuable comments. In the following, we address the comments point by point.
Reviewer 1:
This article provides an in-depth analysis of the treatment pathways and prognosis of advanced sarcoma with peritoneal sarcomatosis. The authors present a comprehensive overview and observation of the clinical evidence. The article is well-written and clearly laid out, making it easy to follow the complex information presented. It is an invaluable resource for clinicians working with advanced sarcoma patients.
Minor revisions to be carried out.
Discuss the potential benefits and drawbacks of different surgical and medical treatments, as well as the role of palliative care.
We have now discussed in our manuscript the main advantages and disadvantages of surgical, medical and also palliative therapy, especially in advanced STS (line 189-217 and line 329-356).
Thorough discussion of the most appropriate treatment strategies should be discussed.
We agree that this information is not included in our manuscript and therefore we have now summarized the current therapeutic approaches for STS in the Discussion section (line 189-217).
The results should also be discussed in the context of clinical approaches for other types of cancer therapeutics referring to latest articles (PMID: 36149747, PMID: 29494958)
Thank you for pointing out these interesting publications. However, we do not find any direct or indirect reference to the treatment of STS in either publication. While one paper deals with the inhibition of HMOX1 ubiquitination in ovarian cancer, the other paper deals with the pharmacological profile of abiraterone acetate in prostate cancer. Alternative therapeutic strategies in terms of targeted therapy and identification of novel molecular targets related to STS have now been discussed in our manuscript based on topic-specific literature (line 205-212).
Reviewer 2 Report
The authors investigate and suggest treatment pathways and prognosis in advanced sarcoma with peritoneal sarcomatosis. The study is interesting, however, I have some concerns to discuss.
-What is the novelty of the current study?
-Please show all patients enrolled in details.
-Please describe the demerit of the tailor-made therapy for sarcomatosis.
Author Response
We thank you for the review of our manuscript and the valuable comments. In the following, we address the comments point by point.
Reviewer 2:
The authors investigate and suggest treatment pathways and prognosis in advanced sarcoma with peritoneal sarcomatosis. The study is interesting, however, I have some concerns to discuss.
-What is the novelty of the current study?
Thank you for bringing this important point to our attention. What is new is that, unlike most studies, we did not focus on a single histologic subtype and did not attempt to group sarcoma patients as a whole. In addition, there is a distinct lack of data focusing on advanced stages such as PS. Furthermore, to the best of our knowledge, no study has focused on the role of surgery as a palliative modality in PS. We have highlighted this in the revised introduction (line 66-69)
-Please show all patients enrolled in details.
We have compiled the detailed characteristics of each patient in a supplementary table.
-Please describe the demerit of the tailor-made therapy for sarcomatosis.
Thank you for raising this important point. In our revised manuscript, we have discussed the disadvantage of tailored therapy, specifically using the example of patients with peritoneal sarcomatosis treated with palliative surgery (line 329-356 and line 363-369)
Round 2
Reviewer 2 Report
The authors replied well ,so the manuscript is suitable for publication.